# STMP-Net: A Spatiotemporal Prediction Network Integrating Motion Perception

**DOI:** 10.3390/s23115133

**Published:** 2023-05-28

**Authors:** Suting Chen, Ning Yang

**Affiliations:** School of Electronic and Information Engineering, Nanjing University of Information Science and Technology, Nanjing 210044, China; 20211249197@nuist.edu.cn

**Keywords:** video prediction, motion perception, spatiotemporal features, contextual attention mechanism

## Abstract

This article proposes a video prediction network called STMP-Net that addresses the problem of the inability of Recurrent Neural Networks (RNNs) to fully extract spatiotemporal information and motion change features during video prediction. STMP-Net combines spatiotemporal memory and motion perception to make more accurate predictions. Firstly, a spatiotemporal attention fusion unit (STAFU) is proposed as the basic module of the prediction network, which learns and transfers spatiotemporal features in both horizontal and vertical directions based on spatiotemporal feature information and contextual attention mechanism. Additionally, a contextual attention mechanism is introduced in the hidden state to focus attention on more important details and improve the capture of detailed features, thus greatly reducing the computational load of the network. Secondly, a motion gradient highway unit (MGHU) is proposed by combining motion perception modules and adding them between adjacent layers, which can adaptively learn the important information of input features and fuse motion change features to significantly improve the predictive performance of the model. Finally, a high-speed channel is provided between layers to quickly transmit important features and alleviate the gradient vanishing problem caused by back-propagation. The experimental results show that compared with mainstream video prediction networks, the proposed method can achieve better prediction results in long-term video prediction, especially in motion scenes.

## 1. Introduction

With the wide application of artificial intelligence technology, video prediction based on deep learning algorithms has become a hot problem for research in this field. Video prediction aims to predict and output future video frames based on past video frames, and is widely used in video interpolation [1], rainfall prediction [2,3], autonomous driving [4,5], action prediction [6], etc. It predicts possible future situations based on existing video content to facilitate the early response and treatment of events, thus effectively avoiding unnecessary losses and achieving desired results. Therefore, as an important research element in video processing and analysis, video prediction is of great importance for the development of society.

Unlike traditional one-dimensional time series problems, video prediction belongs to the category of two-dimensional spatiotemporal sequence problems. For video prediction tasks, the model needs to abstract and extract various detailed information from the images, which not only includes static spatial information, but also dynamic temporal information. Therefore, the key to video prediction lies in effectively capturing the spatiotemporal features and motion changes of video frames. In recent years, deep learning technology has made great achievements in deep feature extraction of multimedia data [7,8,9], and based on this, deep-learning-based video prediction methods have been proposed. In particular, methods based on Recurrent Neural Networks (RNNs) have been proven to be effective for video prediction. As shown in Figure 1, the traditional ConvLSTM [2] network extracts spatiotemporal features layer by layer through multiple stacked ConvLSTMs, and learns long-term and short-term dependencies in the video sequence in the LSTM unit, outputting the predicted result of the next frame.

However, the limited dynamic input sequence and the complex variations of motion itself severely limit the model’s ability to express inter-frame motion information and transform complex spatiotemporal features, and the multi-layered stacking prediction structure can also lead to the disappearance of inter-frame motion features [10]. Therefore, a highly spatiotemporal-correlated video prediction model is needed to accomplish this task. In response to the difficulty of recurrent neural networks in capturing long-term dynamics in context and inter-frame motion changes, this paper proposes spatiotemporal attention fusion units to extract and learn spatiotemporal feature information and improve the ability to capture long-term dynamics. At the same time, a contextual attention mechanism is introduced in this unit [11], which broadens the receptive field and can give different degrees of attention to different temporal states. In order to learn inter-frame motion changes and predict complex motion scenes, this paper proposes motion-aware adaptive connection layers to learn inter-frame motion changes [12], greatly improving the model’s ability to predict motion scenes. Finally, the effectiveness of the model is evaluated by testing on the Moving MNIST dataset [13] and the KTH Human Action Recognition dataset [14]. The experiments show that this method can fully integrate spatiotemporal memory features and motion change features, and comprehensively model the long-term and short-term dynamics of video frames to achieve effective prediction of the future state of the video. The main contributions of this paper are as follows:1.This paper proposes a recurrent neural network-based video prediction method (STMP-Net) combining spatiotemporal memory features and motion perception;2.A spatiotemporal attention fusion unit (STAFU) is proposed based on a gated recurrent unit, which adaptively learns important contextual information through a contextual attention mechanism, and also greatly improves the network’s ability to capture video spatiotemporal features by using a spatiotemporal memory unit;3.Motion perception is introduced based on a gradient highway unit, and a motion gradient highway unit (MGHU) is proposed to learn transient changes and motion trends between video frames. Additionally, a shortcut gradient is provided between adjacent layers to mitigate the gradient disappearance problem caused by backpropagation.

## 2. Related Work

In recent years, research in the field of video prediction has been very active, and new methods and technologies continue to emerge, including traditional methods of video prediction, deep-learning-based video prediction [15,16], and methods that combine motion estimation and multi-scale video prediction.

Traditional video prediction methods typically use techniques based on motion estimation and interpolation, such as optical flow-based methods [17,18,19], which have been proven to be effective in video prediction. Patraucean et al. [20] used a spatiotemporal autoencoder to learn and store the implicit representation of videos, and predicted optical flow information through the collaborative action of the flow prediction module and the spatiotemporal decoder, and applied it to generate future frames. Liu et al. [1] trained a deep convolutional neural network to encode existing video frames and convert them into a voxel flow representation of the video. Then, they used a method based on optical flow to learn the motion trajectory of the video and generate predicted video frames. These methods can predict the content of future video frames, but their predictive performance is poor for complex dynamic scenes and motion patterns.

Deep-learning-based video prediction methods have been a hot research topic in recent years [21,22,23], including methods based on convolutional neural networks (CNNs) and recurrent neural networks (RNNs) [24]. In particular, methods combining long and short-term memory networks (LSTM) provide new ideas for accomplishing video prediction tasks [13,25,26,27]. Shi et al. [2] processed the data using convolutional operations, and achieved weight sharing and pooling operations similar to convolutional operations by stacking convolutional layers and LSTM layers, while also possessing the modeling ability of LSTM for sequence dimensions. Ballas [28] proposed ConvGRU based on ConvLSTM, which uses convolutional operations to set the gate control, enhances parameter sharing capabilities, and focuses on key spatial positions to strengthen the correlation between model units, improving the model’s spatiotemporal modeling ability. Wang et al. [10,29,30] proposed the PredRNN prediction model, which captures the spatiotemporal dependencies of video data by using ST-LSTM that combines spatial and temporal dimensions, thereby obtaining better spatial and temporal features of video data. To solve the problem of gradient vanishing caused by multi-layer deep prediction, GHU and Causal LSTM are introduced, and Predrnn++ is proposed. These methods can learn high-level features of video frames, thus better predicting the content of future video frames, but they are expensive in long-term prediction and complex scene prediction.

In addition, some researchers have proposed methods based on Generative Adversarial Networks (GANs). Liang et al. [31] designed a dual-motion GAN architecture, which requires the predicted future frames to be consistent with the pixel flow in the video through a dual learning mechanism, forming a closed-loop feedback message to improve the prediction performance. Huang et al. [32] used an autoregressive method for single-image video prediction, which preserves more details from the input image while capturing key pixel-level changes between frames. To better predict the content of future video frames, Refs. [14,33,34] combined motion estimation with deep learning techniques. These methods can better capture motion information in videos, thereby improving the accuracy of predictions.

Although the methods mentioned above have made certain progress in video prediction tasks, there are still certain limitations, as shown in Table 1. Especially in long-term video prediction tasks with complex spatiotemporal relationships and fast motion changes, the performance is often unsatisfactory. Therefore, this paper integrates multiple optimization methods and proposes a video prediction network that combines spatiotemporal features and motion perception. Based on LSTM, spatiotemporal storage units and contextual attention mechanisms are introduced, and spatiotemporal attention fusion units are proposed to extract spatiotemporal and detail features from video frames. Based on GHU, a motion perception module is introduced, and a motion gradient highway unit is proposed to learn motion characteristics between video frames and mitigate the effects of gradient vanishing. By coordinating the work of different modules, the prediction performance of the network is greatly improved.

## 3. Model Design

In this paper, we construct a video prediction network with a coding-prediction structure based on a recurrent neural network, which integrates the long- and short-term dynamics of video frames, learns and preserves the inter-frame motion changes, and greatly improves the performance of the prediction network.

### 3.1. Prediction Method

This paper proposes a video prediction method based on Recurrent Neural Networks (RNNs) that combines spatiotemporal memory features and motion perception. The prediction process is shown in Figure 2 (left), which first samples a certain number of video frames from the human behavior recognition video and preprocesses them. Then, the preprocessed video frames are organized into a sequence, and the data in the sequence are standardized and reshaped into a 5D tensor that can be processed by the model. The input sequence can be divided into multiple time steps, and each time step is used as input data for the model. Secondly, a spatiotemporal prediction network, STMP-Net, is constructed by four layers of stacked spatiotemporal attention fusion units and motion gradient highway units between layers to encode and predict the data. Finally, the predicted future frames are reconstructed as output ${\hat{X}}_{1},{\hat{X}}_{2},\ldots ,{\hat{X}}_{t},{\hat{X}}_{t+1}$. 

The network architecture of STMP-Net is shown in Figure 2 (right). From the core architecture of the network, the input data $X_{t}$ are first processed by spatiotemporal attention fusion units with attention mechanisms to extract features from the continuous video frames while learning spatial and temporal feature information $H_{t}^{l}$. The use of attention mechanisms can shift attention to important features and provide a more complete memory of subtle changes in local input sequences, enabling more comprehensive learning of the long-term and short-term dynamic features of video frames [35], reducing resource waste and improving the accuracy of video prediction. Secondly, the long-term information of previous time steps is transmitted to the current time step by horizontally connecting the motion trend $D_{t-1}^{l}$ and the motion change information $T_{t-1}^{l}$ of the previous time step. Then, the short-term motion changes information $D_{t}^{l}$ and $T_{t}^{l}$ between frames are learned and enhanced by the motion gradient highway units and passed to the next time step. By building a fast channel through skip connections between the motion gradient highway units and the spatiotemporal attention fusion units, the input feature $M_{t}^{l}$ and the output feature $Z_{t}^{l}$ are directly added to avoid excessive gradient changes, and normalization is used to further stabilize gradients, effectively alleviating the problem of gradient vanishing. The above operations are repeated to extract features layer by layer, and in the last layer of each time step, the spatiotemporal attention fusion units take the output feature vector as input, and the gradients of the model are calculated by a back-propagation algorithm to update the model parameters. By repeating this process multiple times, future spatiotemporal sequence data are predicted and the output video prediction frame ${\hat{X}}_{t}$ is reconstructed. In addition, since the network structure is independent of layers, the temporal feature information can only be transmitted within each layer. Therefore, a zigzag spatiotemporal information flow $M_{t-1}^{4}$ is introduced to ensure that the top-level temporal features of the previous time step are not lost when extracting features through the spatiotemporal attention fusion units in the next time step.

### 3.2. Motion Gradient Highway Unit (MGHU)

Due to the multilayer stacked prediction structure of recurrent neural networks, the gradient is always multiplied by the same weight in the back-propagation and is prone to the problem of gradient disappearance. It has been shown that introducing adaptable layers between layers can effectively solve this problem, so in this paper, we choose to introduce adaptable connection layers between layers of the prediction network and provide a more efficient route for the gradient to return from the output to the remote input in the past by adding a fast channel. Most of the current prediction models only consider the transition of spatiotemporal states between frames and ignore the internal short-term motion changes leading to the general effectiveness of the model in predicting scenes with complex motions. Therefore, this paper introduces a motion-aware module based on the common highway unit to learn and predict the motion trends and transient changes of subjects and proposes a motion gradient highway unit with the structure shown in Figure 3.

The MGHU calculation process is as follows:(1)$$\begin{array}{c} P_{t}=\tanh\left(W_{p}*\mathrm{Concat}\left(\left[\mathrm{Enc}\left(H_{t}\right),T_{t-1}\right]\right)\right)\\ S_{t}=\sigma \left(W_{S}*\mathrm{Concat}\left(\left[\mathrm{Enc}\left(H_{t}\right),T_{t-1}\right]\right)\right) \end{array}$$
(2)$$\begin{array}{c} D_{t}=D_{t-1}+\alpha \left(T_{t-1}-D_{t-1}\right)\\ T_{t}=T_{t-1}⊗\left(1-S_{t}\right)+P_{t}⊗S_{t}+D_{t} \end{array}$$
(3)$$\begin{array}{c} m_{t}=\mathrm{broadcast}\left(\sigma \left(W_{m}*\mathrm{Enc}\left(H_{t}\right)\right)\right.\\ H_{t}{}^{\prime }=m_{t}⊗\mathrm{wrap}\left(\mathrm{Enc}\left(H_{t}\right),T_{t}\right)\\ g_{t}=\sigma \left(W_{g}*\mathrm{Concat}\left[\mathrm{Dec}\left(H_{t}^{\prime }\right),H_{t}\right]\right)\\ Z_{t}=g_{t}⊗H_{t-1}+\left(1-g_{t}\right)⊗\mathrm{Dec}\left(H_{t}^{\prime }\right) \end{array}$$

Formula (Equation 1) represents the calculation process of GHU. In this process, the spatiotemporal attention fusion units extract spatiotemporal features $H_{t}^{l}$ from the video sequence, which is then encoded and combined with the motion change information $T_{t-1}^{l}$ in the gradient highway module. The gate mechanism controls the selection and retention of features, enabling the adaptive extraction of important information from the input features. $P_{t}$ represents the transformed input, and $T_{t-1}$ represents the motion change feature captured by the filter in the previous time step. $S_{t}$ is a control switch that can perform adaptive learning between the transformed input $P_{t}$ and the motion change feature $T_{t-1}$. $W_{p}$ and $W_{s}$ represent the weights assigned by the convolution filter, and $\mathrm{Enc}\left(H_{t}\right)$ represents the input from the prediction block.

Formula (Equation 2) represents the calculation process of the motion perception module. The motion trend $D_{t-1}^{l}$ and motion change information $T_{t-1}^{l}$ from the previous time step are transmitted to the motion perception module. Through the Sub operation, the motion trend and transient variation $D_{t}^{l}$ of the input features at the current time step are learned. Then, the learned motion information is weighted and fused with the information preserved by the GHU module to obtain the output information $T_{t}^{l}$ at the current time step. α is the step size for momentum updates, and $W_{m}$ represents the weights assigned by the motion filter.

Formula (Equation 3) represents the calculation process of the modulation module. The fused spatiotemporal feature information is passed through the motion filter $m_{t}$ and the tensor dimension is modulated to match the dimension of $H_{t}{}^{\prime }$. Then, the edge pixel values are calculated through the wrap operation with bilinear interpolation to improve the visual representation and visual effect of the image [12]. Finally, the gate mechanism is used to adjust the input and encoded output, controlling the degree of update of the input information to the state, thereby improving the expressiveness and flexibility of the model. $g_{t}$ is an output modulation gate, $\mathrm{Dec}\left(H_{t}{}^{\prime }\right)$ is the decoded output, and $Z_{t}$ is the output feature.

### 3.3. Spatiotemporal Attention Fusion Unit (STAFU)

Since ST-LSTM tends to ignore some detailed information during feature extraction and video prediction, it is necessary to selectively focus on some features to better handle key objects and improve the accuracy of data feature understanding without increasing the computational cost. In this paper, we choose to introduce the contextual attention mechanism [11], and the structure is shown in Figure 4. First, the convolution layer is used to extract the static context among the key points of the input information. Then, the static context is connected with the query and continuously passed through a 1 × 1 convolution with an activation function $W_{\theta }$ and a 1 × 1 convolution without an activation function $W_{\delta }$ to obtain the context attention matrix. Next, self-attention is performed based on the attention matrix and values to obtain dynamic contextual information. Finally, the dynamic contextual information and static contextual information are fused and output.

The attentional mechanism calculation process is shown in Formula (Equation 4): (4)$$\begin{array}{c} A=\left[K^{1},Q\right]W_{\theta }W_{\delta }\\ K^{2}=A*V\\ ATT=\;\mathrm{Fusion}\;\left[K^{1},K^{2}\right] \end{array}$$

*K, Q, V* are keys, queries, and values, $K^{1}$, $K^{2}$ are the static and dynamic contexts of the input information, $W_{\theta }$ and $W_{\delta }$, are the 1 × 1 convolution with and without the ReLU activation function, respectively. A is the contextual attention matrix, and ATT is the output after the fusion of dynamic and static contextual information.

In addition, this article proposes a spatiotemporal attention fusion unit based on the standard gated recurrent unit (GRU), which combines attention mechanism and spatiotemporal memory features, as shown in Figure 5. As an important component of the spatiotemporal prediction network, the innovation of the spatiotemporal attention fusion unit lies in the simultaneous introduction of spatiotemporal information features and attention mechanism, which can learn both the temporal and spatial features in human behavior video frames, as well as focus on the details in the image.

From the internal structure perspective, firstly, the video frame input $X_{t}$ at the current time step and the output $H_{t-1}^{l}$ from the previous time step are selectively attended to by the attention mechanism to better process key objects and improve the accuracy of data feature understanding without increasing computational costs. Secondly, a temporal modulation gate $g_{t}$ is introduced to interact with the reset gate, which controls the amount of information that needs to be retained from past time for the input data of human behavior recognition videos. This improves the model’s ability to capture complex short-term dynamics between adjacent time steps and allows the input data to pass through at different rates adaptively. Then, a set of transformation functions generate the temporal memory unit $T_{M}$, which is the temporary memory transferred from the previous time node to the current time node, and the update gate is used to control the input amount of temporal information at the current state. Additionally, the spatiotemporal memory state $M_{t}^{l-1}$ of the current time step is introduced, and after being processed by the attention mechanism, it transfers the state information across layers and forwards it to the next time step. It not only extends the network’s state transition path in the temporal dimension, but also adds additional storage units between horizontally adjacent nodes at the same level in the spatial dimension, allowing the network to learn complex non-linear transformation functions under short-term motion. To capture the long-term dependencies in the spatiotemporal transformation process, the spatiotemporal memory storage unit $S_{M}$ is defined, which can read and update the memory state based on its understanding of the spatiotemporal state when information transfer passes through this node. Finally, the temporal memory $T_{M}$ and spatiotemporal memory $S_{M}$ are connected and dimensionally reduced by a 1 × 1 convolution layer, and the update gate is used to control the input amount of current time information to achieve collaborative learning of long-term and short-term dynamics.

The calculation process in STAFU is as follows:(5)$$\begin{array}{c} r_{t}=\sigma \left(W_{xr}*X_{t}+W_{hr}*H_{t-1}^{l}\right)\\ g_{t}=\tanh\left(W_{xg}*X_{t}+W_{hg}*H_{t-1}^{l}\right)\\ u_{t}=\sigma \left(W_{xu}*X_{t}+W_{hu}*H_{t-1}^{l}+b_{u}\right)\\ H_{att}=ATT\left(H_{t-i}^{k}+X_{t}\right)\\ T_{M}=\tanh\left(H_{att}⊗r_{t}⊗g_{t}+X_{t}\right) \end{array}$$
(6)$$\begin{array}{c} r_{t}^{\prime }=\sigma \left(W_{xr}^{\prime }*X_{t}+W_{mr}^{\prime }*M_{t}^{l-1}\right)\\ u_{t}^{\prime }=\sigma \left(W_{xu}^{\prime }*X_{t}+W_{mu}^{\prime }*M_{t}^{l-1}+b_{u}^{\prime }\right)\\ M_{att}=ATT\left(M_{t-i}^{k}\right)\\ S_{M}=\tanh\left(M_{att}⊗r_{t}^{\prime }+X_{t}\right) \end{array}$$
(7)$$\begin{array}{c} H_{t}^{l}=\left(u_{t}*T_{M}+u_{t}^{\prime }*S_{M}\right)*\tanh\left(W_{1\times 1}*\left[T_{M},S_{M}\right]\right) \end{array}$$

Formula (Equation 5) is the process of calculating the time memory storage unit, where $X_{t}$ is the input at the current time and $H_{t-1}^{l}$ is the output at the moment t − 1. $r_{t}$ and $g_{t}$ denote the reset gate, modulation gate and update gate, respectively. ATT denotes the attention mechanism, and *W* and *b* denote the weight and bias of the corresponding gating mechanism, respectively.

Formula (Equation 6) is the calculation process of the spatiotemporal memory unit. The input information is controlled by the reset gate and update gate to determine the amount of information to be retained. The spatiotemporal memory feature information, after being processed by the attention mechanism and reset gate, retains key information in the spatiotemporal memory unit $S_{M}$. Here, $r_{t}^{\prime }$ and $u_{t}^{\prime }$ represent the reset gate and update gate of the spatiotemporal memory unit working jointly with the current time input. $W^{\prime }$ and $b^{\prime }$ represent the weights and biases of the corresponding gate mechanisms, respectively. $M_{t}^{l-1}$ is the spatiotemporal memory state at the current time step.

Formula (Equation 7) is the calculation process of the modulation module. The data fused by the temporal memory unit $T_{M}$ and the spatiotemporal memory unit $S_{M}$ are dimensionally reduced and controlled by the update gate to obtain the final hidden state. Here, $u_{t}$ and $u_{t}^{\prime }$ represent the update gates of the temporal and spatiotemporal memory, respectively. $W_{1\times 1}$ is a 1 × 1 convolution layer.

### 3.4. Loss Function

In this paper, a joint loss function [36] is used to train the generative model, which can balance the stability and accuracy of the model at the same time. Among them, $L_{1}$ loss is calculated based on the absolute value error, which can penalize large error samples and has stronger stability for outliers. In contrast, $L_{2}$ loss is more concerned with small error samples and is more sensitive to the overall accuracy of the model.
(8)$$\begin{array}{c} L_{P}=L_{1}+L_{2}=\alpha ∥X-\hat{X}∥+\left(1-\alpha \right)∥X-\hat{X}{∥}^{2} \end{array}$$

The loss function is calculated as shown in Formula (Equation 8), where α is the weight of the $L_{1}$ loss function. In this paper, we choose an adaptive training strategy based on the Adam optimizer to dynamically adjust the weights of $L_{1}$ and $L_{2}$ in the training of the model.

## 4. Experiments and Analysis

In this paper, we choose the Adam optimizer based on the joint loss function for adaptive training and optimization of the model in the experimental part, evaluate our method on two publicly available datasets, Moving MNIST, and KTH human action recognition datasets, and assess the effectiveness of the prediction model using mean square error (MSE), structural similarity index (SSIM), peak signal-to-noise ratio (PSNR), and Linearly-Perceived Image Patch Similarity (LPIPS).

### 4.1. Moving MNIST

The dataset contains two randomly sampled digits from the original MNIST dataset. Each digit is shifted in a random direction in a 64 × 64-pixel grayscale image. The entire dataset has a fixed number of entries, 10,000 sequences for training, 3000 for validation, and 5000 for testing [30]. In this dataset, the future trajectory of moving digits is predictable based on enough historical observations, and the training model in this paper predicts the next ten frames from the first ten frames.

#### 4.1.1. Comparison Experiments

The visual prediction effects of different methods on the Moving MNIST dataset (10 frames → 10 frames) are shown in Figure 6. A total of two more representative cases are selected for analysis in this paper. In the first case, there is no entanglement between the predicted digits, and in the second case, the predicted digits appear entangled. When using ConvLSTM and PredRNN for multi-frame prediction, due to the stacking of multi-layer prediction units and the lack of gradient processing, the multi-frame prediction effect is poor, and the figures appear blurred and partially missing in the face of overlapping figures. MIM and MotionRNN perform better, and only a small number of predicted frames appear slightly blurred locally. The STMP-Net proposed in this paper has significantly improved the clarity and completeness of the predicted frames. Even in the case of digital overlap, clear and complete prediction images can be generated.

Table 2 compares the prediction errors of different methods on the Moving MNIST dataset (10 frames → 10 frames). The proposed STMP-Net achieves a prediction error of 29.3 in MSE, which is a reduction of 71.6% and 14.3% compared to ConvLSTM and MotionRNN, respectively. In terms of SSIM, the prediction result of STMP-Net reaches 0.935, which is an improvement of 0.228 and 0.009 compared to ConvLSTM and MotionRNN, respectively. STMP-Net also shows improvement in both metrics compared to other similar prediction models. A lower MSE and higher SSIM indicate better visual quality.

#### 4.1.2. Ablation Study

To obtain more objective experimental results, a series of ablation experiments are conducted in this paper, and the results are shown in Table 3, where the experiments are based on the same loss function and are trained using the Adma adaptive training strategy.

As shown in Table 3, compared with M1, adding MGHU to the model in M2 reduces MSE by 25% and improves SSIM by 0.034, indicating that the addition of MGHU can effectively improve the model’s fitting effect and prediction quality. Compared with ST-LSTM, the STAFU proposed in this paper reduces MSE by 10% and improves SSIM by 0.013, demonstrating that STAFU has a stronger feature extraction ability, which can effectively improve the model’s prediction accuracy and image clarity. A comparison of the prediction effects is shown in Figure 7. In a comprehensive view, the prediction error of the model in this paper is much smaller than that of the benchmark model, and the prediction quality is significantly improved.

### 4.2. KTH Human Action Recognition Dataset

The KTH human action recognition dataset contains six different human actions: walking, jogging, running, boxing, waving, and clapping, in which 25 people perform the actions in four different environments. On average, each video lasts 4 s and is further divided into 25 frames per second. We adjusted the video resolution of individual frames to 128 × 128, where the acting performances of 1–16 people were used as training; the acting performances of the 17th–25th people were used as testing, and a total of 108,717 sequences were obtained for the training set and 4086 sequences for the testing set [30].

#### 4.2.1. Ablation Experiment

To further verify the performance of the spatiotemporal attention fusion unit and motion gradient highway unit in complex motion change scenes, this paper builds using four different prediction models: M1, M2, M3, and ours, by changing the basic structure of the backbone network, and conducts prediction experiments on the KTH human behavior recognition dataset (10 frames → 20 frames), respectively, and the prediction effect is shown in Figure 8. From the figure, it can be seen that ST-LSTM is slightly less effective in prediction due to the lack of effective gradient loss reduction and the ability to perceive short-term motion changes, with blurrier images and gradual loss of local details of the person in the long-term prediction process. In contrast, STAFU incorporates the attention mechanism, which can improve the understanding of spatiotemporal features and detail features, and MGHU enhances the model’s learning of motion change features. The model in this paper has significantly improved the prediction quality of human motion videos, especially the moment of motion change, and the detailed prediction of the motion is clearer.

The quantitative results of the ablation experiments are shown in Table 4. Compared to M1 and M2, MGHU improves PSNR by 5.4% and SSIM by 3.9%, indicating that MGHU significantly enhances the network’s ability to learn motion change features and thus achieves better prediction accuracy and visual effects. Compared to M1 and M3, the STAFU proposed in this paper can better capture spatiotemporal features, with a PSNR improvement of 2.3% and an SSIM improvement of 1.7% compared to the traditional ST-LSTM unit. Overall, our prediction network has better prediction performance than the baseline network, with an improvement of 7.1% in PSNR and 5.8% in SSIM.

#### 4.2.2. Comparison Experiments

To verify the actual prediction effect of the method, this paper chooses to compare the KTH dataset (10 frames → 20 frames) with the mainstream methods of today, and the comparison of the prediction effect is shown in Figure 9.

Figure 9 shows the prediction results of a random sample of an example from the test set by the current mainstream methods, and the comparison shows that the ConvLSTM model generates blurred images with some missing action details during the prediction process. Conv-TT-LSTM, PredRNN, and PredRNN-V2 are slightly better in short-term prediction, but the predicted frames appear blurred in long-term prediction. The prediction frames appear significantly blurred in long-term prediction. Compared with the above methods, the STMP-Net model proposed in this paper can obtain more stable images, especially in the first period, and the images can be output clearly and completely, but the later prediction frames have slight blurring, but it can also meet the prediction requirements and realize the long-term prediction of motion video.

Experiments were conducted on the KTH human action recognition dataset, and the models were evaluated using three metrics, PSNR, SSIM and LPIPS, where the quantitative results of different methods on the KTH dataset (10 frames → 20 frames) are shown in Table 5. Compared with the base PredRNN, the STMP-Net model proposed in this paper has improved 0.48 in PSNR, SSIM by 0.056 and LPIPS decreased by 0.047. It indicates that the model proposed in this paper has higher prediction performance and can generate predicted images with higher clarity.

## 5. Comparative Analysis

In order to further analyze the advantages and limitations of STMP-Net, we compared and analyzed it with current mainstream video prediction methods.

Models such as DVF [1], PredCNN [40], and DPG [41] use convolutional neural networks for video prediction, which have simple structures but often require complex training strategies. Models such as ConvLSTM [2], PredRNN [29], PredRNN++ [10], and MIM-LSTM [37] use stacked RNNs for video prediction, which improve the feature extraction ability of the model by improving the basic module of RNN, but their performance for long-term prediction is poor. Models such as E3D-LSTM [36], CrevNet [38], and PhyDNet [42] use an encoding–prediction–decoding structure for video prediction, which effectively improves the prediction performance of the model by improving the LSTM and encoder–decoder modules, but they perform poorly in complex motion scenarios.

In addition, methods such as [17,18,19,20] are based on optical flow for video prediction, which have good prediction performance for low-texture areas and moving objects, but have strict conditions and are easily affected by factors such as background and motion speed. Methods such as [31,32] are based on GAN for video prediction, which have higher prediction performance and stability, but have long training and prediction times and require high computing resources.

In summary, compared with deep learning prediction models based on CNN and RNN, STMP-Net has significant advantages in long-term prediction and complex motion scenarios. However, its prediction performance in simple scenarios may be slightly inferior to that of convolutional neural network models. Compared with optical flow and GAN, STMP-Net, as a video prediction model based on recurrent neural networks, has higher universality. In addition, the model structure and training method of STMP-Net are relatively simple, but a large number of model parameters may lead to overfitting and longer training time.

## 6. Conclusions

This paper proposes a video prediction method based on recurrent neural networks that combines spatiotemporal memory features and motion perception to effectively predict future frames of a video. The spatiotemporal attention fusion unit is used to learn the long-term and short-term dynamic features between multiple frames. The attention mechanism can adaptively allocate weights, reduce computational load, and improve prediction performance. The fast channel transmission of information between layers avoids the gradient vanishing problem caused by multi-layer stacked prediction. Meanwhile, the motion perception module is introduced to learn the motion changes between frames, greatly improving the prediction accuracy. Experimental results show that STMP-Net achieves PSNR, SSIM, and LPIPS of 31.7, 0.887, and 0.092, respectively, on the KTH dataset, which outperforms current mainstream models, indicating that STMP-Net has good performance in long-term prediction of videos, especially in complex motion scenarios.

## 7. Future Directions

Video prediction is one of the important research directions in the field of computer vision. The current mainstream methods are still based on deep learning methods, but these prediction methods still have certain limitations. In the face of long-term prediction, multimodal prediction and other tasks, the prediction effect is poor; therefore, we believe that future research can be carried out in the following directions:1.Long-term prediction: Most of the current video prediction methods are based on short-term prediction, and future research directions can explore how to better perform long-term prediction, i.e., predicting tens or even a hundred frames in the future. This will have greater requirements on the complexity of the model and the amount of data.2.Multimodal video prediction: The current video prediction mainly considers the data of image sequences, and if other modal data can be integrated, such as audio, semantic segmentation, etc., there will be better prediction results. Future research directions can explore how to integrate data from different modalities for video prediction to improve the accuracy and robustness of prediction.3.Adapt to different scenes: Current video prediction models are usually trained and predicted in specific scenes, and often do not perform as well in the face of different scenes or different motion patterns. In the future, we can explore how to design pervasive video prediction models that can adapt to different scenes and motion patterns.

## Figures and Tables

**Figure 1 sensors-23-05133-f001:**
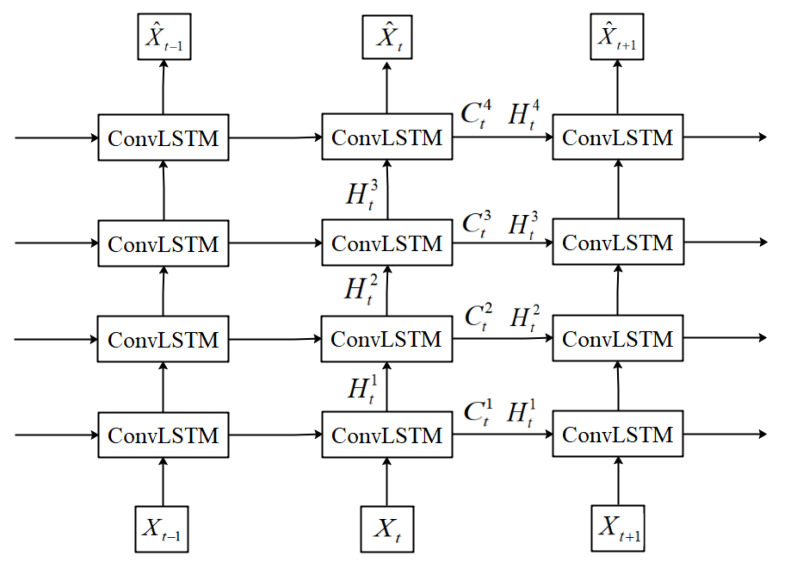
Traditional ConvLSTM network architecture.

**Figure 2 sensors-23-05133-f002:**
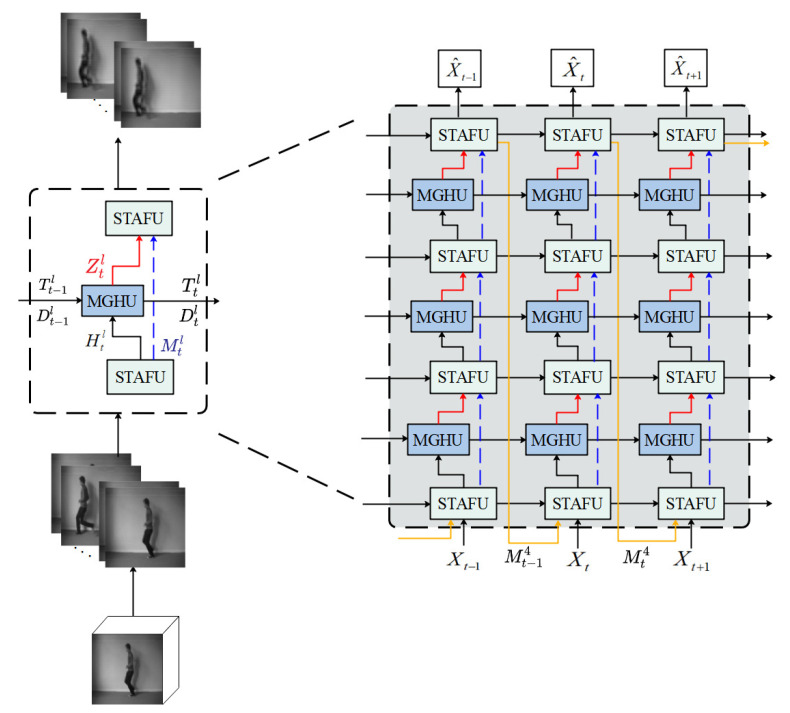
(**Left**) Flow chart of video prediction method combining spatiotemporal memory features and motion perception. (**Right**) the main architecture of STMP-Net.

**Figure 3 sensors-23-05133-f003:**
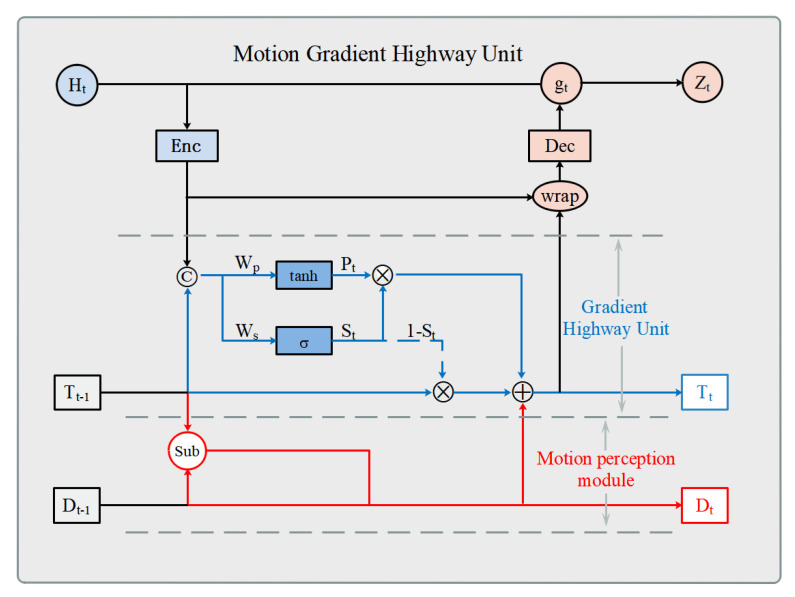
Schematic diagram of the structure of the motion gradient highway unit.

**Figure 4 sensors-23-05133-f004:**
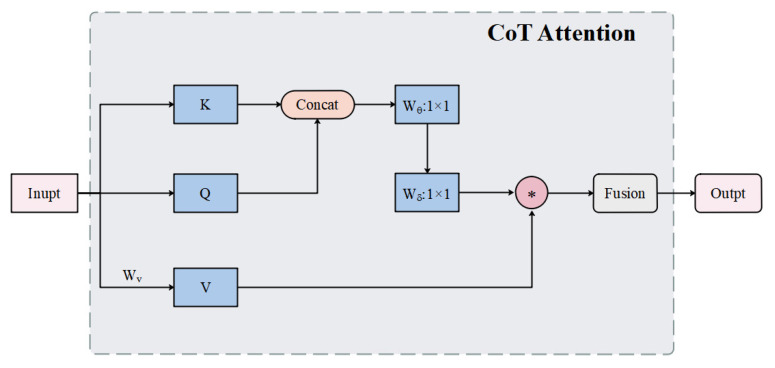
Schematic diagram of the structure of the contextual attention mechanism.

**Figure 5 sensors-23-05133-f005:**
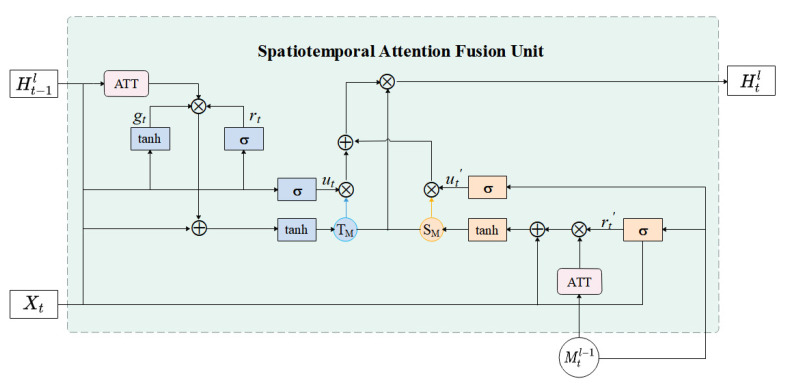
Schematic diagram of the structure of the spatiotemporal attention fusion unit.

**Figure 6 sensors-23-05133-f006:**
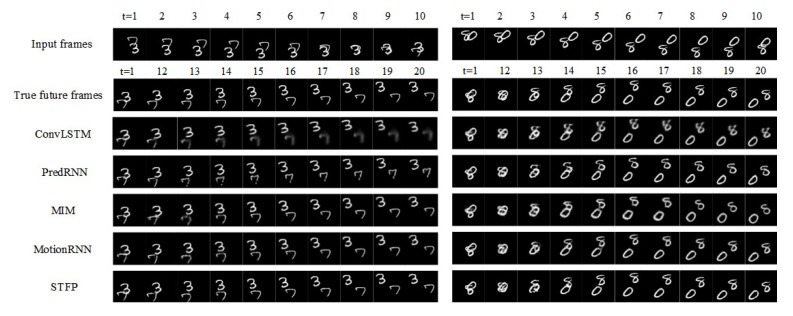
Visual prediction results of different methods on the Moving MNIST dataset (10 frames → 10 frames).

**Figure 7 sensors-23-05133-f007:**
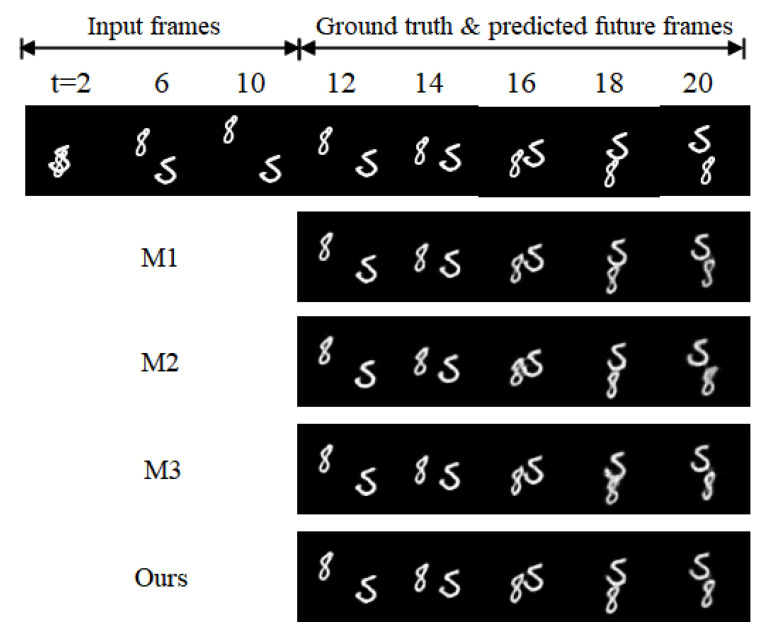
Effect diagram of ablation experiment prediction.

**Figure 8 sensors-23-05133-f008:**
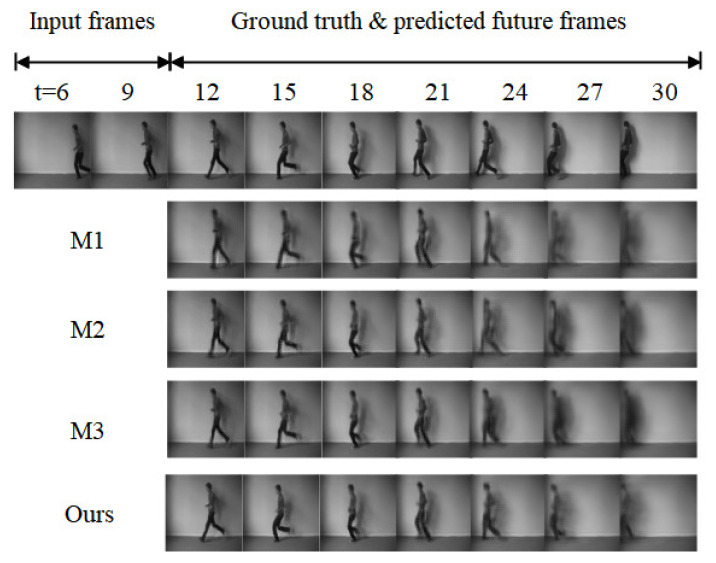
Predicted effect of the ablation experiment.

**Figure 9 sensors-23-05133-f009:**
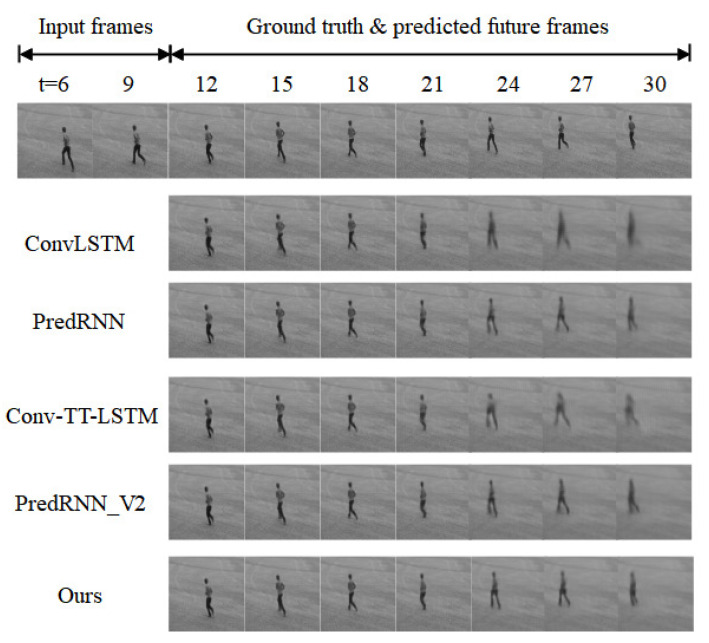
Comparison of prediction effects of different models on KTH dataset (10 frames → 20 frames).

**Table 1 sensors-23-05133-t001:** Limitations of related methods.

Methods	Related Literatures	Limitations
Traditional methods	[1,17,18,19,20]	The prediction accuracy is affected by factors such as video resolution and frame rate.
Deep Learning methods	[2,10,13,24,25,26,27,28,29,30]	Poor performance in long-term prediction and high time cost for both training and inference.
Motion estimation methods	[14,31,32,33,34]	High requirements for both the quantity and quality of data and motion estimation.

**Table 2 sensors-23-05133-t002:** Quantitative results of different methods on the Moving MNIST dataset (10 frames → 10 frames).

Model	MSE (↓)	SSIM (↑)
ConvLSTM [2] (2015)	103.3	0.707
PredRNN [29] (2017)	56.8	0.867
MIM [37] (2019)	44.2	0.910
CrevNet [38] (2020)	38.5	0.928
MotionRNN [12] (2021)	34.2	0.926
STMP-Net	29.3	0.935

**Table 3 sensors-23-05133-t003:** Ablation study of the Moving MNIST dataset (10 frames → 10 frames).

Model	Backbone	MSE (↓)	SSIM (↑)
M1	4×STLSTM [30]	47.5	0.889
M2	4×STLSTM, 3×MGHU	35.2	0.923
M3	4×STAFU	42.7	0.902
Ours	4×STAFU, 3×MGHU	29.3	0.935

**Table 4 sensors-23-05133-t004:** Ablation study of the KTH human action recognition dataset (10 → 20).

Model	Backbone	PSNR (↑)	SSIM (↑)
M1	4×STLSTM [30]	29.6	0.838
M2	4×STLSTM, 3×MGHU	31.2	0.871
M3	4×STAFU	30.3	0.852
Ours	4×STAFU, 3×MGHU	31.7	0.887

**Table 5 sensors-23-05133-t005:** Quantitative results of different methods on the KTH dataset (10 frames → 20 frames).

Model	PSNR (↑)	SSIM (↑)	LPIPS (↓)
ConvLSTM [2]	24.6	0.753	0.212
PredRNN [29]	28.4	0.832	0.198
Conv-TT-LSTM [39]	27.3	0.825	0.172
PredRNN-V2 [30]	29.6	0.838	0.139
Ours	31.7	0.887	0.092

## Data Availability

Not applicable.

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
