# Peer review of "STMP-Net: A Spatiotemporal Prediction Network Integrating Motion Perception"

_sensors, 2023, doi:10.3390/s23115133_

Round 1

Reviewer 1 Report

1. Extend abstract and add some more information

2. Add one related figure in introduction section and explain main contribution points

3. Add limitation table in related work section

4. Explain in detail Prediction model and explain figure 1 as well

5. Formula 1,2 and 3 need to be properly explained 

6. Explain figure 3 and 4 

7. Add separate section of future directions 

8. Add another section of theoretical comparative analysis and add atleast 15 update papers also show comparative analysis with proposed solution 

Overall paper need to be update in terms of english writing 

Reviewer 2 Report

General Comments: The manuscript investigates and proposes a video prediction network STMP-Net integrating motion perception and spatiotemporal memory. this is a useful topic and the article provides a clear study of the technical concepts. and the method development as well as the results are clearly presented. However, it still needs to be improved.

Comments 1:Several figures, tables, and equations have been inserted in the main text, but are not properly referenced in the appropriate places in the text; the references need to be completed.

Comments 2: Figure 1 contains the method flowchart and network structure schematic for video prediction, but the current caption of Figure 1 illustrates the text is not clear and complete. I believe that a more detailed description should be provided in the caption of Figure 1.

Comment 3:The explanation of equations (6), (7) in Section 3.3 is too simple and lacks the description of relevant parameters to provide effective help. I think a more detailed explanation should be added.

Comment 4: In Section 4.2.1, the authors only briefly summarize the analysis of the quantitative results of the ablation experiments in Table 3, which I believe should be analyzed and explained in more detail based on the experimental results.

1.  The English needs to be revised throughout. The authors should pay attention to the spelling and grammar throughout this work. I would only respectfully recommend that the authors perform this revision or seek the help of someone who can aid the authors.

2. (Section 1 Introduction) The reviewer hopes the introduction section in this paper can introduce more studies in recent years. The reviewer suggests authors don't list a lot of related tasks directly. It is better to select some representative and related literature or models to introduce with certain logic. For example, the latter model is an improvement on one aspect of the former model.

Reviewer 3 Report

This study proposed a recurrent neural network (RNN)-based video prediction method called  STMP-Net. The main elements of STMP-Net are the 'Spatio-Temporal Attention Fusion Unit (STAFU)' and 'Motion Gradient Highway Unit (MGHU)'. The authors claim that this method captures temporal changes in the video more effectively and mitigates the gradient loss problem caused by backpropagation. Thus, STMP-Net is designed to capture temporal motion and change, as well as spatial features, simultaneously and integrate them to have the ability to effectively predict the future state of the video. This reviewer questioned the following points:

(1) The KTH human action recognition dataset is a dataset containing relatively simple behaviours and the performance of the model for complex real-world scenarios has not been tested. It would need to be evaluated on more complex datasets and real-world scenarios.

(2) The proposed model contains complex components, such as spatio-temporal feature extraction and a motion perception module. These components may require a high level of computational power and therefore may not be practical in resource-limited environments. How much time and computer resources are required to operate at a practical level?

(3) Deep learning models that take into account temporal dependencies are often difficult to train and are particularly prone to problems such as the gradient disappearance problem. This paper solves this problem using a motion gradient highway unit, but deep learning models are generally regarded as "black boxes" and their predictions can be intuitively difficult to understand. It is difficult to understand the inner workings of a proposed model and why it produces good results.

(4) Do the authors plan to make this AI model available to the public for testing? It is difficult to assess whether it is really better than other methods, as the authors claim, based only on the benchmark comparisons presented in this paper.

Round 2

Reviewer 1 Report

Still this point need to be addressed 

Add a section of comparative analysis and add at least 15 updated papers to compare it with proposed solution 

Overall paper need to be proofread properly 

Reviewer 2 Report

The paper needs to undergo minor revisions. Some comments are as follows.

1. The font size of the text in the figures is too small, which affects the reader's reading.

2. Details on the experiments. The results should be discussed and analyzed in detail.

3. The references should have a unified format.

Overall, I would encourage the authors to resubmit it after careful modification.

The paper needs to undergo minor revisions. Some comments are as follows.

1. The font size of the text in the figures is too small, which affects the reader's reading.

2. Details on the experiments. The results should be discussed and analyzed in detail.

3. The references should have a unified format.

Overall, I would encourage the authors to resubmit it after careful modification.

Reviewer 3 Report

This paper has been improved by the revisions. Also, as many, if not all, of the questions have been answered, I would support the acceptance of this paper for Sensors.

Author Response

Thank you for your valuable input on our paper, which led to its successful publication in Sensors.